# Nitric oxide mediates neuro-glial interaction that shapes *Drosophila* circadian behavior

**Anatoly Kozlov**, **Rafael Koch**, **Emi Nagoshi***

Department of Genetics and Evolution, Sciences III, University of Geneva, Quai Ernest-Ansermet, Switzerland

* Emi.Nagoshi@unige.ch

## Abstract

*Drosophila* circadian behavior relies on the network of heterogeneous groups of clock neurons. Short- and long-range signaling within the pacemaker circuit coordinates molecular and neural rhythms of clock neurons to generate coherent behavioral output. The neurochemistry of circadian behavior is complex and remains incompletely understood. Here we demonstrate that the gaseous messenger nitric oxide (NO) is a signaling molecule linking circadian pacemaker to rhythmic locomotor activity. We show that mutants lacking nitric oxide synthase (NOS) have behavioral arrhythmia in constant darkness, although molecular clocks in the main pacemaker neurons are unaffected. Behavioral phenotypes of mutants are due in part to the malformation of neurites of the main pacemaker neurons, s-LNvs. Using cell-type selective and stage-specific gain- and loss-of-function of NOS, we also demonstrate that NO secreted from diverse cellular clusters affect behavioral rhythms. Furthermore, we identify the perineurial glia, one of the two glial subtypes that form the blood-brain barrier, as the major source of NO that regulates circadian locomotor output. These results reveal for the first time the critical role of NO signaling in the *Drosophila* circadian system and highlight the importance of neuro-glial interaction in the neural circuit output.

## Author summary

Circadian rhythms are daily cycles of physiological and behavioral processes found in most organisms on our planet from cyanobacteria to humans. Circadian rhythms allow organisms to anticipate routine daily and annual changes of environmental conditions and efficiently adapt to them. Fruit fly *Drosophila melanogaster* is an excellent model to study this phenomenon, as its versatile toolkit enables the study of genetic, molecular and neuronal mechanisms of rhythm generation. Here we report for the first time that gasotransmitter nitric oxide (NO) has a broad, multi-faceted impact on *Drosophila* circadian rhythms, which takes place both during the development and the adulthood. We also show that one of the important contributors of NO to circadian rhythms are glial cells that form the blood-brain barrier. The second finding highlights that circadian rhythms of higher organisms are not simply controlled by the small number of pacemaker neurons but are generated by the system that consists of many different players, including glia.

**Data Availability Statement:** All relevant data are within the manuscript and its Supporting Information files.

**Funding:** This work was supported by the research grant to EN from the Swiss National Science

Foundation (31003A_169548). AK was partially supported by the Plan Strategique Sciences Vie (PSVIE) of the University of Geneva. The funders had no role in study design, data collection and analysis, decision to publish, or preparation of the manuscript.

**Competing interests:** The authors have declared that no competing interests exist.

## Introduction

Our environment undergoes daily fluctuations in solar illumination, temperature, and other parameters. Organisms across the phylogenetic tree are equipped with circadian clocks, which help predict daily environmental changes and create temporal patterns of behavioral and physiological processes in concordance with the environmental cycle. *Drosophila melanogaster* remains a powerful model to study this phenomenon ever since Konopka and Benzer identified the first clock gene, *period*, in this organism [1].

*Drosophila* circadian clocks rely on transcriptional-translational feedback loops that operate using an evolutionarily conserved principle. In the main loop, CLOCK/CYCLE (CLK/CYC) heterodimers bind to the E-boxes in the promoter regions of the *period* (*per*) and *timeless* (*tim*) genes and activate their transcription. PER and TIM proteins undergo post-translational modifications and enter the nucleus to suppress their own production by inhibiting CLK/CYC activity. CLK/CYC also activates transcription of the genes encoding the basic-zipper regulators PAR DOMAIN PROTEIN 1 (PDP-1) and VRILLE (VRI), which activates and inhibits *Clk* gene expression, respectively. Thus, positive- and negative- feedback loops created by PDP-1 and VRI with CLK/CYC are interlocked with the main negative-feedback loop and ensure the generation of 24-h rhythms [2, 3].

In the fly brain, molecular clocks are present in ca.150 so-called clock neurons, which form the pacemaker circuit controlling circadian behavior. Clock neurons are classified into groups according to their morphological characteristics and location: small and large lateral ventral neurons (s- and l-LNvs), lateral dorsal neurons (LNds), lateral posterior neurons (LPNs) and three groups of dorsal neurons (DN1s, DN2s, DN3s) [4, 5]. Although all clock neurons express a common set of clock genes, they are heterogeneous in terms of neurotransmitter/neuropeptide phenotype, function, and composition of the molecular clock. Neuropeptide pigment-dispersing factor (PDF) is uniquely secreted from the l-LNvs and 4 out of 5 s-LNvs. Several other neuropeptides, including small neuropeptide F (sNPF) and ion transport peptide (ITP), and classical neurotransmitters such as glutamate and glycine, are also expressed across pacemaker circuit [6, 7]. PDF-positive s-LNvs are designated as the Morning (M) oscillator, whereas LNds together with the PDF-negative 5th s-LNv consist of the Evening (E) oscillator. Under the light-dark (LD) experimental conditions, the M and E oscillators drive the morning and evening anticipatory increments of locomotor activity, respectively. The M oscillator is also the master pacemaker of the free-running locomotor rhythms in constant darkness (DD) [8–11].

Neuropeptide PDF as well as the unique composition and regulatory mechanisms of the molecular clock underlie the distinct role of the M oscillator. The main negative-feedback loop of the M oscillator's molecular clock employs a specific phosphorylation program that regulates the nuclear translocation of PER/TIM complex [12]. The nuclear receptor UNFULFILLED (UNF) is almost uniquely present in the lateral neurons within the circadian circuit [13, 14]. UNF accumulates rhythmically in the s-LNvs and, in cooperation with another nuclear receptor E75, enhances CLK-dependent *per* transcription. Thus, UNF and E75 consist a positive limb of an additional feedback loop in specific to the s-LNv molecular clock. Because UNF and E75 also play critical roles in the development of the s-LNvs, knockdown of either gene during development or adulthood results in low rhythmicity and extended period, respectively [14, 15].

Nuclear receptors (NRs) are a superfamily of proteins that function as ligand-dependent transcriptional regulators [16]. The ligands are small lipophilic molecules that can diffuse across the cell membrane, such as thyroid and steroid hormones. In *Drosophila melanogaster*, only two lipophilic hormones, 20-hydroxyecdyson (20E) and the sesquiterpenoid juvenile hormone (JH) are known nuclear receptor ligands, which have critical roles in developmental

processes, including molting, puparium formation, and neurogenesis [16–18]. Although many NRs remain orphan without a known ligand, diatomic gases—nitric oxide (NO) and carbon monoxide (CO)—can bind and regulate the activity of some NRs. Several studies have demonstrated *in vitro* and *in vivo* that NO binds to E75 and regulates its interaction with DHR3 [19], SMRTER [20], and UNF [21] in different tissues during development. Thus, the binding of NO to E75 confers an important switching mechanism in various developmental processes.

NO is an unconventional messenger involved in numerous biological functions, including immune defense, respiration, intracellular signaling and neurotransmission [22, 23]. NO can act locally near the source of its production. It can diffuse across membranes and also act as a long-range signaling molecule [22, 24]. NO signaling is broadly classified into the classical pathway mediated by cGMP and cGMP-independent non-classical one involving diverse mechanisms such as posttranslational modifications and transcriptional regulations [22, 25]. In mammals, the importance of NO signaling in the light-dependent phase-resetting and maintenance of rhythmicity [26–28] is established. These effects were largely explained by the canonical NO/cGMP signaling [29–31]. However, whether NO has a regulatory role in *Drosophila* circadian behavior has never been addressed.

Here we explore the role of NO in circadian locomotor behavior of *Drosophila* using multiple genetic approaches. We present evidence that NO signaling is necessary for proper circadian locomotor behavior. NO can act cell-autonomously as well as non-cell-autonomously at multiple processes required for generating rhythmic behavior, including axonal morphogenesis, phasing of molecular clocks and output control. Furthermore, we identify the perineurial glia, one of the two glial subtypes that form the blood-brain barrier, as a source of NO that controls free-running locomotor output. Our results highlight the complexity of locomotor behavior regulation and oft-neglected importance of glia in the regulation of behavior.

## Results

### *dNOS* deletion mutants show arrhythmic circadian behavior

NO is chiefly produced by an enzyme nitric oxide synthase (NOS) through the conversion of arginine into citrulline using NADPH as a cofactor [32, 33]. Three distinct NOS isoforms (endothelial e-NOS, inducible i-NOS, and neuronal n-NOS) exist in mammals, whereas *Drosophila* has a single *NOS* (*dNOS*) gene that produces 10 splice variants (S1 Fig) [34]. Since NOS functions as homodimers, alternatively spliced variants, most of which encode truncated proteins, are proposed to act as dominant negatives [35]. To investigate the possible roles of NO in fly circadian rhythms, we took advantage of two *NOS* CRISPR deletion mutants, *NOSΔall*, and *NOSΔter* [21]. The former has a deletion of the entire NOS locus, while the latter is a partial deletion mutant lacking exons 1 to 6 but bears intact two uncharacterized genes within the *NOS* locus (Fig 1A). RT-qPCR using the primers targeting the exons commonly included in all variants (exons 10 and 11) confirmed the absence of the full-length NOS1 mRNA expression in *NOSΔall* mutants (Fig 1B) A reduced level of the product was detected in *NOSΔter* mutants, consistent with the location of the deletion. Furthermore, we directly measured NO production in cultured whole brains using a fluorescent dye DAR4-M [36]. NO was virtually undetectable in both *NOSΔall* and *NOSΔter* strains in this assay, confirming that both are complete loss-of-function mutants (Fig 1C).

Having validated the *NOS* CRISPR deletion mutants, we next tested their locomotor activities in LD and DD paradigms. Homozygous mutants had strongly reduced rhythmicity in DD. Trans-heterozygous of two deletion alleles was equally detrimental to DD rhythmicity, whereas heterozygous mutations had no effect on rhythmicity (Table 1 and Fig 2). Moreover, morning activity patterns in LD were strongly impaired in homozygous and trans-heterozygous

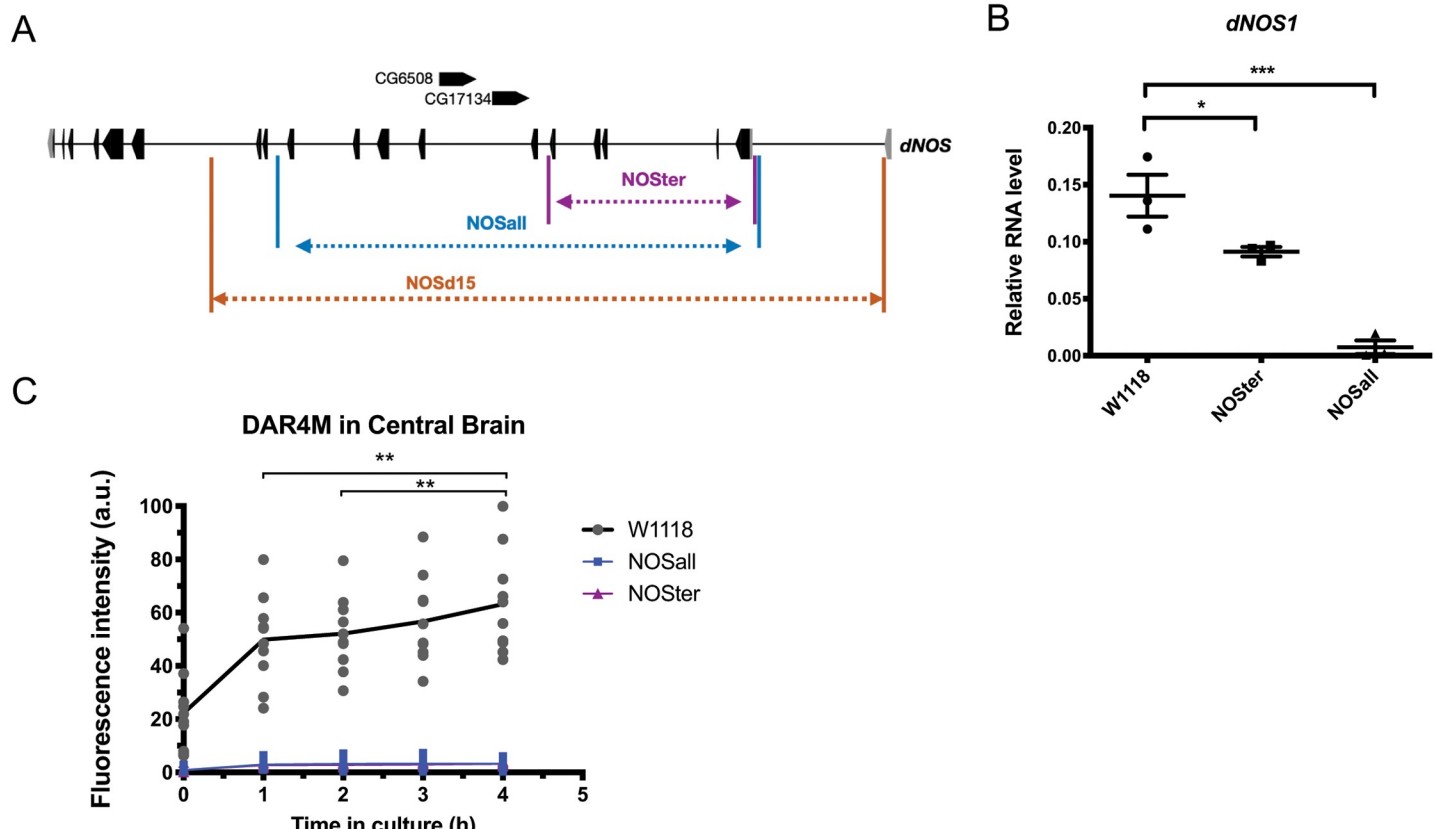

**Fig 1. *NOSΔ* mutants do not produce nitric oxide.** (A) *NOS* gene, and regions deleted in *NOSΔall*, *NOSΔter* and *NOS^{Δ15}* mutants. Two genes reside within the *NOS* locus. (B) mRNA levels of *NOSd1*, a full-length functional isoform of *NOS*, at ZT2 in the heads of *NOSΔall* and *Δter* mutants and *w^{1118}* were analyzed using qPCR. Mean values of three independent experiments ± SD are shown. *NOSd1* expression levels were significantly reduced in the mutants compared with *w^{1118}*. $^*p<0.05$, $^{***}p<0.001$ (One-way ANOVA with Dunnett's multiple comparisons test). (C) NO levels were measured using DAR4-M dye in the brain explants of *NOSΔall* and *Δter* mutants and *w^{1118}* for 4 h using timelapse microscopy. Each dot represents the DAR4-M fluorescence intensity of a single brain explant and the line indicate the mean in arbitrary unit (a.u.). n = 5–11 brains per group. Different time points within the group were compared using two-way ANOVA with Tukey's multiple comparisons test ($^{**}p<0.01$).

mutants. In particular, the DD-arrhythmic flies lack both anticipatory increments of activity before lights-on (Fig 2, right). Since two *NOS* CRISPR*Δ* mutants share a common genetic background, we also tested locomotor behavior of the well-characterized *NOS^{Δ15}* mutant produced by recombination between two piggyBac transposon insertions [37] (Fig 1A) and *NOS^{Δ15}* heterozygous with a chromosomal deficiency (*Df(2L)BSC230*) that deletes numerous genes including *NOS*. NOS enzymatic activity is completely disrupted in *NOS^{Δ15}* mutant [37]. The locomotor behavior of *NOS^{Δ15}* homozygous and hemizygous mutants in DD was similar to those of *NOS* CRISPR deletion mutants, showing strongly reduced rhythmicity (Table 1 and Fig 2). LD behavior was also impaired in these mutants. Although the startle response to lights-on was present, *NOS^{Δ15}* homozygous and hemizygous mutants had reduced morning anticipatory activities. Taken together, these results demonstrate that *NOS* is required for normal circadian locomotor activity rhythms.

## NOS is a regulator of morphogenesis of the s-LNv axons

NO signaling plays critical roles in various developmental processes in the nervous system, including neurite patterning of the visual system and axon pruning/regrowth of mushroom body (MB) neurons [19–21, 38, 39]. Since low rhythmicity in DD and poor morning

**Table 1. Effects of NOS mutation, NOS knockdown, and macNOS overexpression on free-running locomotor rhythms.** The left-most column explains the types of experiments and where the drivers are expressed. *n*, number of flies. %R, % of rhythmic flies. CTR, control. Oxp, over-expression. KD, knockdown. OP, optic lobes. Mean periods and rhythmicity of the test groups were compared with those of control groups using Student's t-test and chi-square test, respectively. Mutants were compared with *Canton-S* and flies for knockdown and overexpression were compared with GAL4-only controls.

| | Genotype | Period ± SEM (hr) | Power ± SEM | n | %R |
|---|---|---|---|---|---|
| **Mutants and CTR** | $w^{1118}$ | 23.5±0.05 | 167.5±9.3 | 124 | 93.6 |
| | *Canton-S* | 24.1±0.35 | 102.3±18.0 | 27 | 84.4 |
| | *NOSter/+* | 23.7±0.04 | 217.7±10.5 | 127 | 91.3 |
| | *NOSter* | 23.2±0.2 | 84.3±29.0 | 105 | 21.0*** |
| | *NOSall/+* | 23.7±0.04 | 245.1±12.1 | 126 | 76.9 |
| | *NOSall* | 23.6±0.06 | 149.4±15.5 | 173 | 56.1** |
| | *NOSall/NOSter* | 23.5±0.2 | 79.0±17.4 | 59 | 66.1 |
| | *NOSdelta15* | 23.6±0.08 | 70.2±8.00 | 31 | 45.2** |
| | *NOSdelta15/+* | 23.5±0.09 | 148.3±9.27 | 31 | 87.1 |
| | *Df/+* | 23.6±0.07 | 118.8±9.33 | 29 | 89.7 |
| | *NOSdelta15/Df* | 24.0±0.05 | 101.5±9.57 | 32 | 65.6* |
| **macNOS oxp** | *macNOS/+* | 23.4±0.05 | 59.4±2.1 | 89 | 85.4 |
| **LNvs** | *PDF>macNOS* | 24.5±0.15*** | 51.7±6.1 | 25 | 76.6 |
| **s-LNVs** | *R6>macNOS* | 23.8±0.09 | 69.9±5.2 | 31 | 65.5** |
| **MB** | *D52H>macNOS* | 24.2±0.05 | 145.2±13.9 | 28 | 92.9 |
| **Clock neurons** | *tim>macNOS* | 24.7±0.1**** | 61.3±6.3 | 30 | 36.7**** |
| **Clock neurons** | *Clk1982>macNOS* | 23.9±0.05 | 85.5±7.6 | 30 | 76.7 |
| **OL** | *GMR33H10>macNOS* | 23.7±0.09 | 85.2±8.7 | 62 | 53.2**** |
| **OL** | *GMR79D04>macNOS* | 23.9±0.09 | 43.3±4.9 | 63 | 28.6**** |
| **OL** | *GMR85B12>macNOS* | 23.6±0.1 | 53.1±7.9 | 60 | 61.7*** |
| **Glia** | *Repo>macNOS* | 23.5±0.05 | 98.2±5.0 | 28 | 29.8**** |
| **Pan-neuronal** | *GMR57C10>macNOS* | 25.0±0.4* | 65.9±12.9 | 61 | 32.8**** |
| **Pan-neuronal** | *Elav>macNOS* | 23.7±0.05 | 83.5±6.4 | 59 | 81.4 |
| **NOS KD** | *NOS-RNAi²⁷⁷²⁵/+* | 23.7±0.04 | 110.7±0.04 | 145 | 84.0 |
| **LNvs** | *Pdf>NOS-RNAi* | 24.4±0.03 | 184.1±13.9 | 26 | 100 |
| **s-LNVs** | *R6>NOS-RNAi* | 23.6±0.04 | 113.0±5.5 | 60 | 80 |
| **MB** | *D52H>NOS-RNAi* | 23.6±0.4 | 105.4±10.5 | 29 | 65.5* |
| **MB** | *OK107>NOS-RNAi* | 23.5±0.05 | 141.2±12.0 | 32 | 96.9 |
| **Photoreceptors** | *GMR>NOS-RNAi* | 23.6±0.06 | 200.1±12.3 | 60 | 95.2 |
| **Clock neurons** | *tim>NOS-RNAi* | 24.4±0.2 | 166.2±10.4 | 47 | 38.3**** |
| **Clock neurons** | *Clk1982>NOS-RNAi* | 23.6±0.02 | 173.3±5.03 | 62 | 88.7 |
| **OL** | *GMR33H10>NOS-RNAi* | 23.5±0.03 | 164±7.1 | 32 | 96.9 |
| **OL** | *GMR79D04 >NOS-RNAi* | 25.1±0.2** | 105.2±7.9 | 59 | 61.0** |
| **OL** | *GMR85B12 >NOS-RNAi* | 23.6±0.03 | 121.3±6.9 | 44 | 70.5** |
| **Glia** | *Repo>NOS-RNAi* | 23.6±0.2 | 79.4±6.6 | 63 | 32.1**** |
| **Pan-neuronal** | *GMR57C10>NOS-RNAi* | 26.2±0.2**** | 121.2±8.2 | 60 | 55.0** |
| **Pan-neuronal** | *elav>NOS-RNAi* | 23.5±0.05 | 122.3±9.1 | 95 | 76.8 |
| **CTR** | *PDF/+* | 24.1±0.04 | 132.7±6.9 | 58 | 96.8 |
| | *R6/+* | 23.4±0.05 | 105.2±5.7 | 45 | 85.1 |
| | *D52H/+* | 23.9±0.04 | 177.3±8.3 | 10 | 90.5 |
| | *OK107/+* | 23.7±0.06 | 139.6±8.5 | 30 | 96.8 |
| | *GMR/+* | 23.6±0.03 | 187.5±6.1 | 28 | 93.3 |
| | *Tim/+* | 24.1±0.05 | 125.4±6.3 | 82 | 90.2 |
| | *Clk1982/+* | 23.6±0.03 | 150.6±7.4 | 60 | 86.7 |
| | *GMR33H10/+* | 23.1±0.05 | 144.9±6.2 | 60 | 85.0 |
| | *GMR79D04/+* | 24.3±0.2 | 96.6±5.5 | 64 | 84.4 |

*(Continued)*

**Table 1.** (Continued)

| | Genotype | Period ± SEM (hr) | Power ± SEM | n | %R |
|---|---|---|---|---|---|
| | *GMR85B12/+* | 23.5±0.04 | 128.5±6.9 | 91 | 89.0 |
| | *Repo/+* | 23.3±0.05 | 109.2±6.9 | 83 | 82.0 |
| | *GMR57C10/+* | 24.2±0.2 | 117.9±4.9 | 60 | 78.3 |
| | *Elav/+* | 23.7±0.05 | 143.0±7.8 | 120 | 82.8 |
| **18˚C ->29˚C** | *Repo>NOS-RNAi* | 24.1±0.5 | 47.2±7.3 | 21 | 47.6* |
| **Adult-only KD** | *GMR79D04>NOS-RNAi* | 24.7±0.3 | 93.1±8.3 | 28 | 75.0 |
| | *GMR57C10>NOS-RNAi* | 26.8±0.07*** | 165.3±11.5 | 25 | 96.0 |
| **CTR** | *NOS-RNAi/Tub-Gal80ts* | 25.6±0.3 | 89.2±6.5 | 29 | 79.3 |
| | *Repo/Tub-Gal80ts* | 23.5±0.6 | 114.2±9.3 | 29 | 72.4 |
| | *GMR79D04/Tub-Gal80ts* | 23.3±0.07 | 76.0±8.7 | 28 | 67.9 |
| | *GMR57C10/Tub-Gal80ts* | 23.2±0.06 | 121.7±9.74 | 30 | 86.7 |

*$p < 0.05$

**$p < 0.01$

*** $p < 0.001$

****$p<0.0001$.

anticipation are indicative of the dysfunction of the s-LNvs, we sought to examine possible effects of *NOS* deficiency on the structure and function of the s-LNvs. To this end, we expressed a membrane-targeted yellow fluorescent protein mCD8::VENUS with the

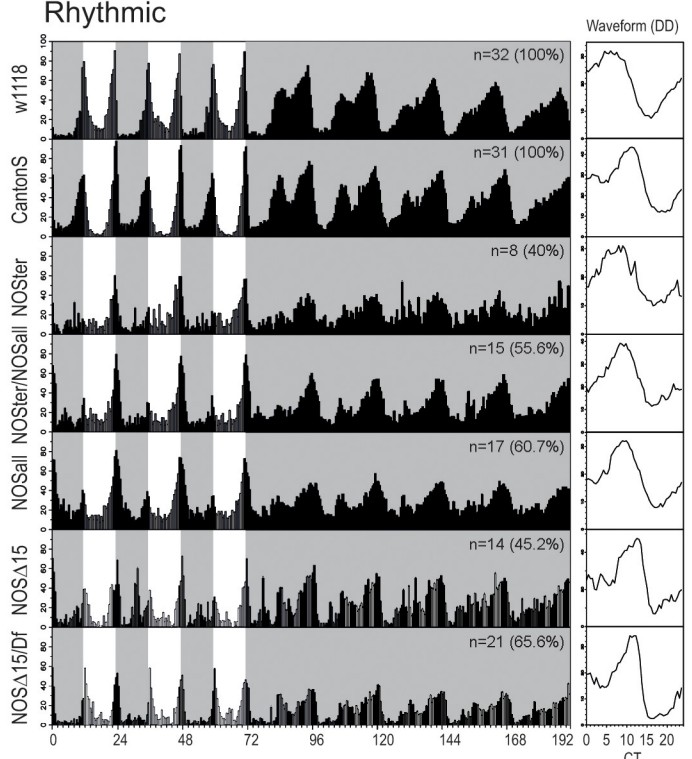
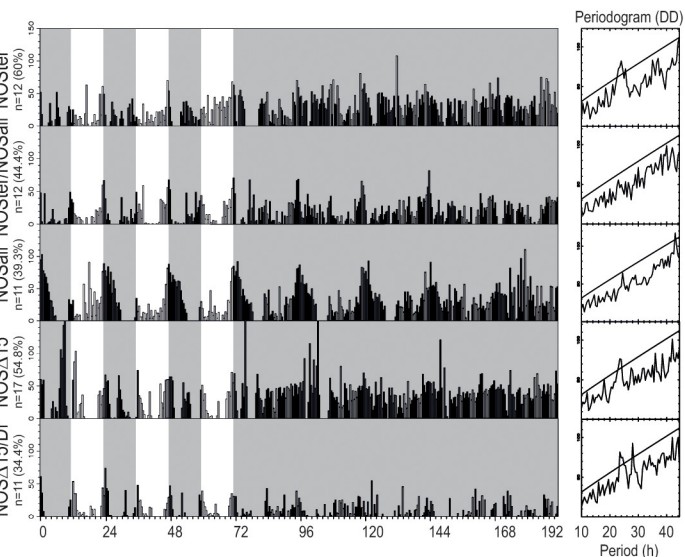

**Fig 2. *NOS* deletion impairs the circadian behavior in constant darkness.** The locomotor activity of *w[1118]*, *Canton-S* and *NOS* homozygous, transheterozygous and hemizygous mutants in LD and DD. Left, group-average locomotor activity and mean waveforms of rhythmic flies. Right, representative actograms and periodograms demonstrating arrhythmic locomotor activity of NOS mutant flies.

*gal1118-Gal4* driver and inspected s-LNv axonal morphology at two timepoints in LD. The levels and rhythms of PDF neuropeptide accumulation in the s-LNvs were also analyzed by immunohistochemically. Surprisingly, branching pattern of terminal neurites of the s-LNvs, which is normally orderly, was severely disturbed in *NOS* homozygous mutants (Fig 3A). The neurites were extended in length and branching pattern was highly disordered and fuzzy, showing the tendency of axon misrouting. The quantification of the pixels covered by the axonal termini confirmed that s-LNv axonal arbors were overgrown in *NOS* mutants (Fig 3C). Furthermore, PDF levels were increased and had no rhythms in the mutants (Fig 3A and 3B). Of note, no statistical difference in axonal areas was detected between ZT2 and ZT14 both in the mutants and the control genotype (Fig 3C). Thus, this quantification method was probably not sensitive enough to detect daily structural changes in this genetic background, although the same method successfully detected rhythmic changes in axonal structure in our previous study [40]. Nevertheless, these observations suggest that NOS deficiency leads to defects in the morphology of s-LNvs dorsal projections and disturbances in the rhythms of PDF accumulation. The latter finding suggests that clocks in the s-LVs are arrhythmic or their downstream events are impaired.

To assess whether molecular clocks are functional in *NOS* mutants, we performed around-the-clock immunostaining of a key clock component PER on the third day of constant darkness (DD3). Surprisingly, neither the phase nor the amplitude of the PER rhythms in the s-LNvs was altered in *NOS* mutants (Fig 3D). Molecular clocks of the LNds also maintained high-amplitude 24-h rhythms in mutants (Fig 3E). Therefore, the arrhythmic behavioral phenotype of *NOSΔ* mutants is uncoupled from the state of the molecular clocks and principally caused by the impaired clock output. The impairments in clock output may be due to the malformation of s-LNvs dorsal projections, which likely disrupt network communication to and from the s-LNvs, in addition to defects in possibly many other cells involved in locomotor output.

## NO from diverse cellular groups can modulate the state of the molecular clocks and behavioral output

Whereas NOS is undoubtedly important for developmental processes, the fact that NO continues to be produced in the brains of adult flies (Fig 1C) suggests that NO may also have an active role in the regulation of circadian rhythms. It was previously shown using an anti-NOS serum [41] that NOS is expressed almost everywhere in the brain. However, since the anti-NOS serum does not distinguish various NOS isoforms, sites of NOS expression may not be identical to the loci of active NO production. Additionally, classical histochemical studies of NADPH diaphorase activity of NOS and soluble guanylate cyclase (sGC)/cGMP immunohistochemistry have suggested that NOS is active in sensory pathways including visual system, in memory circuits including the calyx of the mushroom body, in the central complex, and also in some glial cells [32, 39, 42–44]. Since these were all indirect assessments of NO production, we analyzed the localization of NO in the brain using the NO-specific fluorescent probe DAR4-M. DAR4-M staining showed distinct patterns of cell bodies and neurites in many areas. The signal intensity was particularly high within and around the central complex and in the optic lobe, with cell bodies arranged in concentric semicircles, reminiscent of the laminar and medulla glial cells [45] (Fig 4A). We also detected DAR4-M signal co-localized with surface glia marked by GFP driven with the pan-glial driver *Repo-GAL4* (Fig 4B). These patterns were overall similar to those described for the localization of NADPH diaphorase activity and cGC/cGMP. In addition, since DAR4-M staining is not sensitive enough to assess daily variation of NO production, we examined the temporal expression pattern of the functional

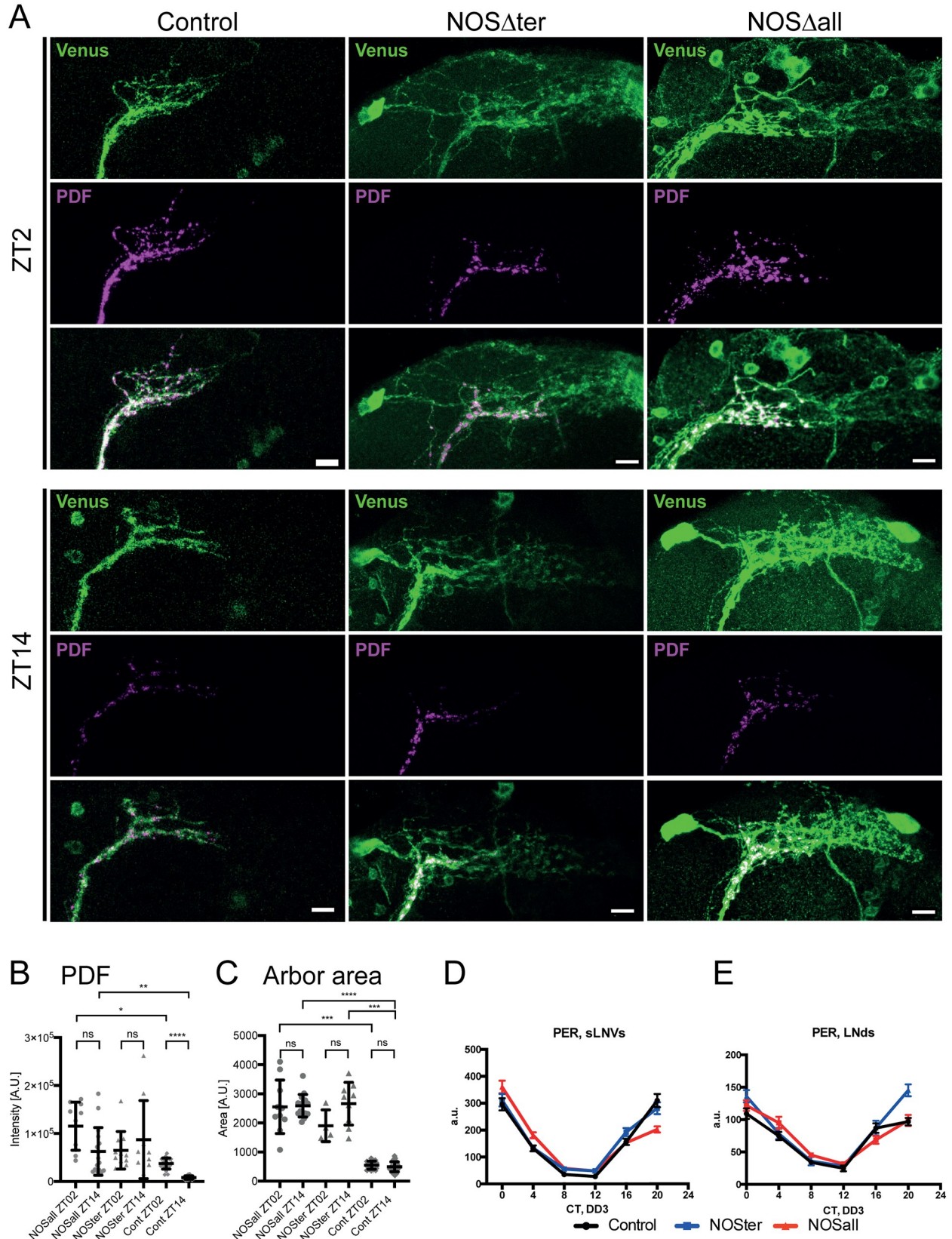

**Fig 3. *NOS* deletion causes the malformation of the axons of the s-LNvs but does not affect their molecular clocks.** (A) s-LNv axonal terminal projections and PDF neuropeptide in *NOS* mutants were visualized by expressing *mCD8::Venus* with *gal1118-GAL4* and staining with anti-GFP and anti-PDF antibodies at ZT2 and ZT14. The control is *gal1118 > mCD8:: Venus* in $w^{1118}$ background. Representative confocal images are shown. For visualization purposes only, the contrast of ZT14 magenta channel (PDF) is enhanced. Scale bar, 10 μm. (B and C) Quantifications of PDF levels in axonal arbor area (B) and the area of the terminal branches. Each dot represents the measurement from one hemisphere. Lines represent mean and SD. Differences across all groups were compared using one-way ANOVA with Brown-Forsythe and Welch test for multiple comparisons. Differences between groups were considered significant if $p<0.05$ (*$p<0.05$, **$p<0.01$, and ****$p<0.0001$). (D and E) PER levels in the s-LNvs (D) and LNds (E) in the control and *NOSΔ* mutants were analyzed every 4 h on DD3 by immunostaining using anti-PER antisera. Values represent mean PER immunofluorescence intensity. Error bars, SEM. NOS deletion does not affect PER rhythms.

isoform *dNOS1* using qPCR. We found that mRNA of the *dNOS1* was rhythmically expressed in the fly head, peaking around ZT10 in LD (Fig 4C), suggesting that overall NO levels in the brain may exhibit circadian variation at least in LD.

Interestingly, we detected a slight enrichment of DAR4-M fluorescence within the s-LNvs marked with *Pdf>mCD8::GFP* (Fig 4D). This was unexpected because previous transcriptome studies found very little or no *NOS* expression within the s-LNvs [13, 46, 47]. This finding also suggests that NO produced elsewhere migrates to the s-LNvs. Within the s-LNvs, transcriptional regulation by E75 and UNF is a unique and important node of the molecular clockwork [14, 15]. Intriguingly, it was shown that heterodimerization of E75 and UNF is inhibited by NO *in vivo* and *in vitro* [21]. Therefore, we asked whether the state of the molecular clocks can be modulated by increasing NO within the s-LNvs. To this end, we overexpressed a macrophage-derived constitutively active NOS (macNOS) under the UAS control [19] using *Pdf-Gal4* and performed PER staining on DD3 every 4 h (Fig 5A). The increase of NO within the PDF positive neurons was confirmed by DAR4-M staining (S2 Fig). This manipulation lead to a delay of the PER induction phase by about 4 h without dampening the amplitude of PER rhythms. In contrast, PER levels and oscillations in the LNds and DN1s showed no change with the overexpression of macNOS in the LNvs (Fig 5B).

*Pdf>macNOS* flies had a slight extension of free-running period. Their circadian rhythmicity appeared slightly reduced but was not significantly different compared with the driver-only control. Because *Pdf-GAL4* is expressed in both s- and l-LNvs, these phenotypes may be a compound effect from both cell types. When we expressed macNOS under the s-LNv-specific *R6-Gal4* driver, we observed a reduction in rhythmicity but no differences in the period length (Table 1). Morning anticipation was blunted in *R6>macNOS* but was apparently normal in *Pdf>macNOS* flies, suggesting that the genetic background rather than macNOS expression in the s-LNvs affected the LD behavior (S3 Fig). Therefore, the major behavioral consequence of forced NO production in the s-LNvs is a reduction in free-running rhythmicity. This is probably a consequence of the misalignment of molecular phases between the s-LNvs and other clock neurons (Fig 5A and 5B).

The results of the DAR4-M staining and the finding that NO can regulate the state of the molecular clocks prompted us to investigate whether NO produced in specific cell types or brain area is important for normal circadian locomotor activity. Therefore, taking into account that NO can act both locally and remotely, we selected a set of GAL4 drivers and drove the expression of macNOS. Locomotor activity of these flies was assayed in standard LD-DD conditions.

As summarized in Table 1, we used two clock cell-specific drivers *Tim-GAL4* and *Clk1982-GAL4*; a mushroom body-specific driver *DH52-GAL4*; three generic optic lobe-specific drivers *GMR33H10-*, *GMR79D04-*, and *GMR85B12-GAL4* (S1 Table); a glia-specific driver *Repo-GAL4*; and two pan-neuronal drivers *elav-GAL4* and *R57C10-GAL4*. Strikingly, all of them except *elav-GAL4* induced a reduction of rhythmicity when driving *UAS-macNOS*. This is probably because pan-neuronal *elav-GAL4* is a weaker driver than another pan-

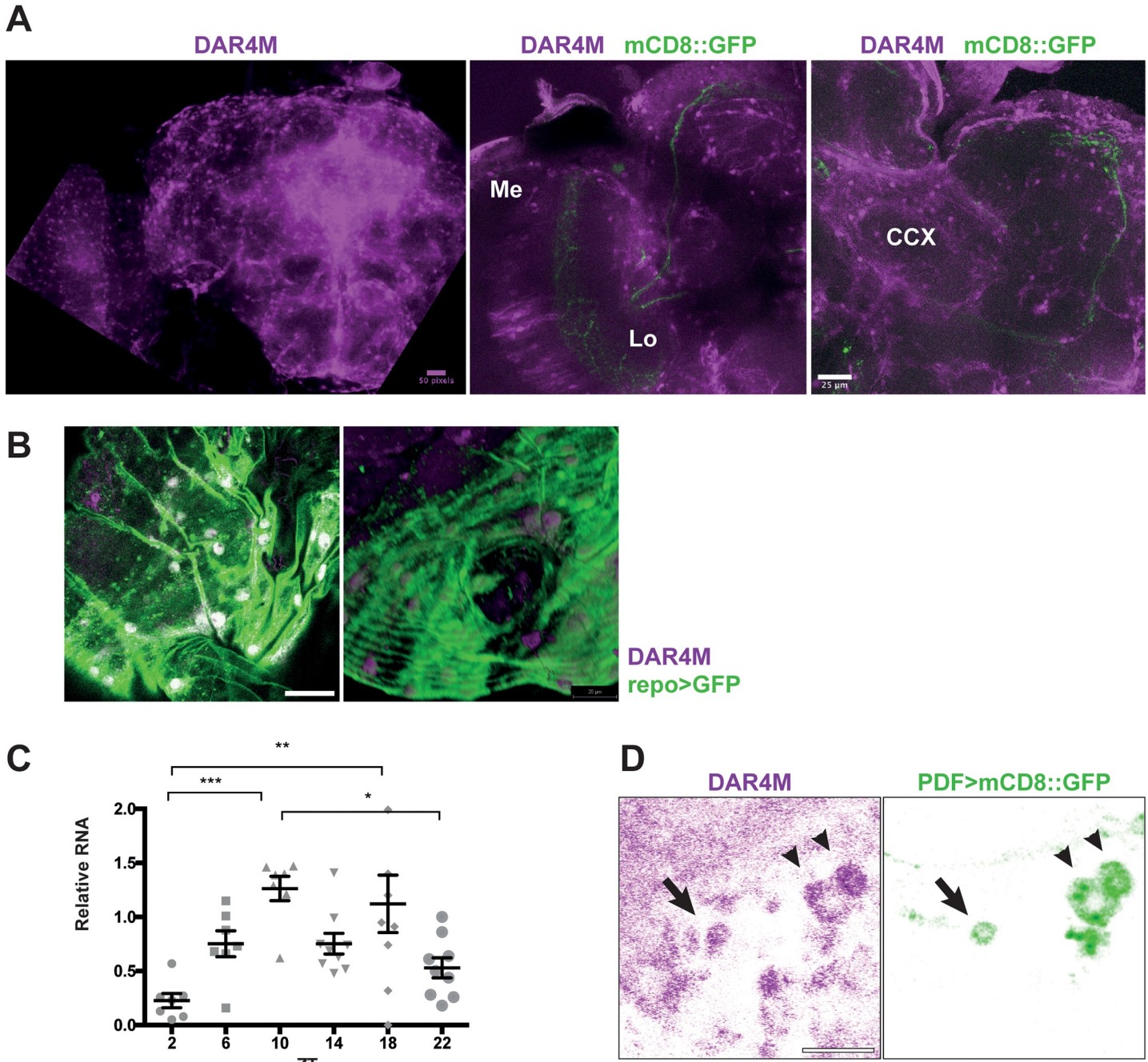

**Fig 4. NO production in the brain.** (A) NO staining with DAR4-M showed an accumulation of NO in the medulla neuropil (Me), lobula (Lo), the central complex (CCX) and its surrounding area. Representative images of the brains of $w^{1118}$ flies (left) and of *Pdf>mCD8::GFP* flies (right) are shown. (B) DAR4-M staining colocalizes in the nuclei of surface glia in the brain marked with *Repo-GAL4>GFP*. Left, a close-up of a single confocal z-plane focusing on the surface glia is shown. Right, a 3D reconstruction of 14 z-sections showing a frontal part of a brain, visualized from a ventral-lateral angle. Scale bar: 20 μm. (C) Relative expression of *dNOS1* isoform in the brain was measured by qPCR in 6 timepoints in LD. Data are shown as mean ± SEM. One-way ANOVA with Dunnett's multiple comparisons test showed significant differences across timepoints (*$p<0.05$, **$p<0.01$, ***$p<0.001$). (D) Enrichment of the DAR4-M signal in the cell body of LNvs marked with *Pdf>mCD8::GFP*. Arrowheads indicate l-LNvs and an arrow indicates a s-LNv. Scale bar, 20 μm. A representative confocal image taken at ZT0 is shown.

neuronal driver *R57C10-GAL4* [48]. The reduction of rhythmicity was markedly dramatic with *Repo*, *tim* and optic lobe specific drivers. LD behavior was not obviously affected in any

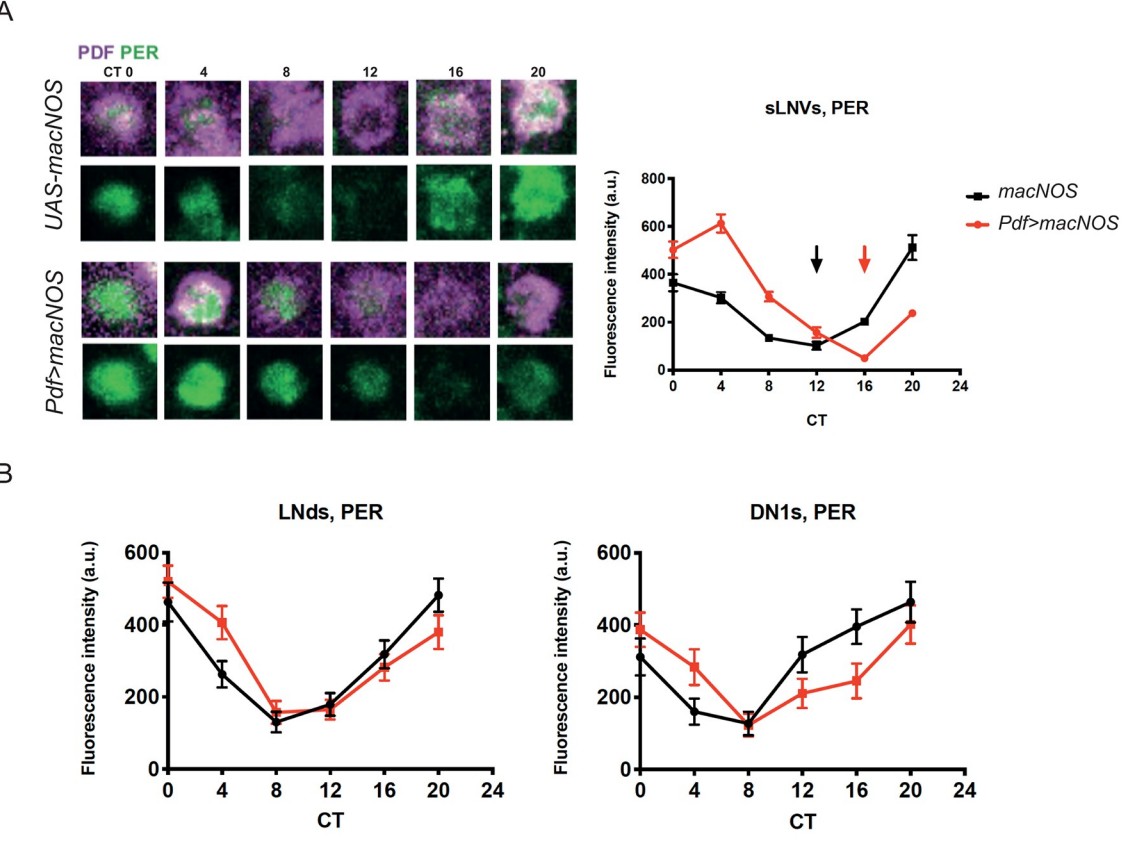

**Fig 5. Hyperproduction of NO delays PER rhythms in the s-LNvs.** PER in the brains of *PDF>macNOS* and control flies carrying only *UAS-macNOS* were monitored every 4 h on DD3 by immunostaining with anti-PER and anti-PDF antibodies. (A) Representative confocal images of the s-LNvs (left) and PER levels in the s-LNvs (right). (B) PER levels in DN1s and LNds. Mean ± SEM. Red and black arrows in (A) point at the trough of PER rhythms in *Pdf > macNOS* and control flies, respectively. PER rhythms in the s-LNvs are delayed by approximately 4 h in *PDF>macNOS* flies.

genotype (S3 Fig). Collectively, these results indicate that overproduction of NO is generally disruptive to locomotor rhythms and suggest that the NO production and clearance should be tightly regulated.

To find cell types that natively produce and secrete NO and contribute to the control of locomotor rhythms, we next performed an opposite experiment. We expressed RNAi against NOS (VDRC #27725) using a similar set of drivers and analyzed its effects on behavioral rhythms (Table 1). Consistent with the likely absence of NOS within the s-LNvs, NOS RNAi with *Pdf-GAL4* and *R6-GAL4* did not show any behavioral phenotype. NOS RNAi driven with a mushroom body driver *DH52-GAL4* caused a reduced rhythmicity, whereas *macNOS* expression with the same driver had no effect. Most of the other drivers that disrupted rhythms with *macNOS* expression also reduced behavioral rhythmicity with *NOS* RNAi. These include a pan-neuronal driver *R57C10-Gal4*, optic-lobe drivers *GMR79D040-Gal4* and *GMR85B12-Gal4*. The strongest effect was observed with *Repo-GAL4* and *tim-Gal4*, whereas there was no effect with *Clk1982-Gal4*. Since the expression of *Clk1982-Gal4* is relatively restricted to CLK-positive neurons, these results suggest TIM-positive glial cells as an important source of NO in the regulation of circadian locomotion. *Repo-Gal4* driving a second independent RNAi against NOS (TRiP #50675) also reduced free-running rhythmicity (3% rhythmic, period 24.0 ± 0 h, n = 31). Another VDRC RNAi line (#108433) had no effect on

behavior with any driver, which is most likely due to an inefficient knockdown compared to the VDRC #27725 line, judging from the NO staining intensity (S4 Fig). LD behavior was differently affected by NOS knockdown with variety of drivers, such as an increase in nighttime activity and lack of the startle response to lights-off. However, all groups showed bimodal activity patterns (S3 Fig). Altogether, the results of NOS gain- and loss-of-function mini screens indicate that NO produced in many different cell types, excluding pacemaker neurons, contribute to generating normal free-running locomotor rhythms.

## NO produced in glia plays an active role in the regulation of locomotor output

Constitutive NOS knockdown may cause structural abnormalities in the brain that lead to the reduction of rhythmicity, as evidenced by the phenotypes of *NOSΔ* mutants (Figs 2, 3A and 3B). Therefore, to test if NOS is required for active maintenance of rhythmicity after eclosion, we performed the adult-specific knockdown of NOS using pan-neuronal, optic lobe specific and glial GAL4 drivers combined with the temperature-sensitive GAL4 repressor, GAL80^ts [49] (Table 1). Glia-specific NOS knockdown caused a notably strong reduction of rhythmicity. In addition, NOS RNAi driven by the pan-neuronal *R57C10-Gal4* extended the free-running period. These results indicate an indispensable role of NO produced in glia for generating circadian locomotor output in adult flies, as well as an existence of a neuronal circuit through which NO signaling regulates the free-running period. PDF levels and rhythms in the s-LNvs were not altered by NOS knockdown with *Repo-GAL4* restricted to adulthood (Fig 6A). The morphology of axonal arbors of the s-LNvs was also not affected by NOS knockdown in adult glia, judging from the distribution of PDF (Fig 6A). These results suggest that glial NOS regulates locomotor rhythms by acting on the process downstream of the circadian pacemaker circuit.

Glia are found throughout the nervous system and play diverse roles, including the maintenance of neurotransmitter and ionic homeostasis, maintenance of the blood-brain barrier, and serving as immune cells. Generic glial subtypes in *Drosophila* are identified as astrocyte-like, cortex, ensheathing, subperineurial and perineurial glia, and can be separately manipulated using specific GAL4 drivers [45]. To examine whether NO produced in specific subtypes of glia regulates circadian locomotor behavior, we drove NOS RNAi with glial subtype-specific drivers and analyzed the locomotor behavior (Table 2). NOS knockdown in perineurial glia with the *GMR85G01* driver markedly reduced locomotor rhythmicity in DD, without affecting LD behavior (Fig 6B and 6C). This phenotype is comparable to that caused by pan-glial NOS knockdown. NOS knockdown with four other drivers resulted in a slight change in period without deteriorating rhythm strength. We therefore conclude that the perineurial glia is the major site of NOS activity that regulates locomotor output. Our DAR4-M staining of NO in surface glia supports this finding (Fig 4B). Additionally, a previous transcriptome study has demonstrated the expression of the *NOS* gene within surface glia [50], further supporting our results. The perineurial glia form the outer layer of the blood-brain barrier, the structure crucial for chemoprotection and selective transport of nutrients (Fig 6D) [51]. Since perineurial glia contain circadian clocks [52], these results are congruent with the arrhythmic behavior caused by NOS knockdown with *Tim-GAL4* (Table 1).

## Discussion

Gaseous signaling molecules play important roles in a myriad of biological processes, including circadian rhythms in mammals. Here we investigated the possible involvement of NO in

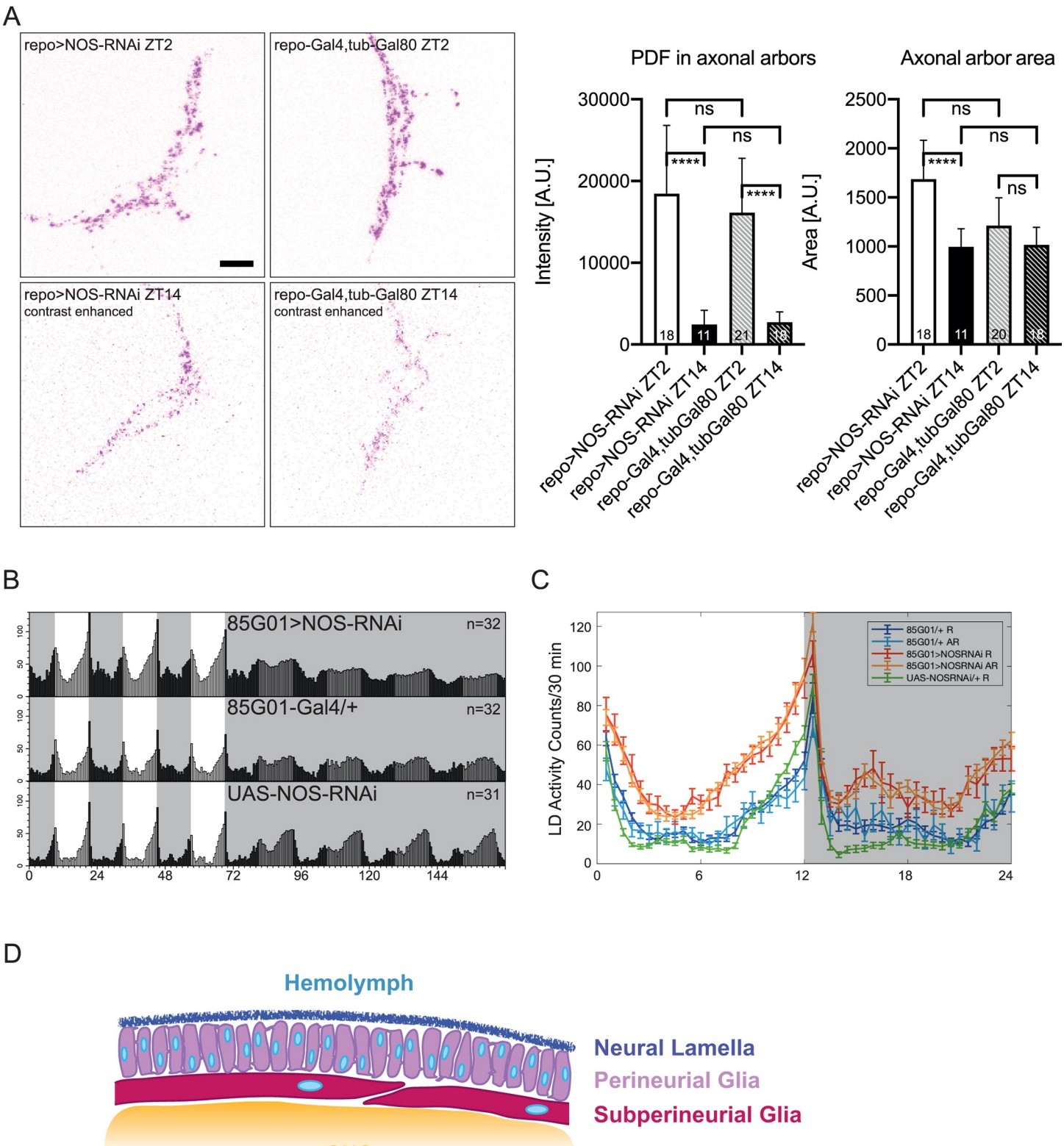

**Fig 6. NO produced in the perineurial glia controls circadian locomotor output.** (A) *NOS RNAi* was driven in glia only during adulthood with *Repo-GAL4, tub-GAL80ts* and a temperature shift from 18˚C to 29˚C after eclosion. Immunostaining with anti-PDF antibody performed at ZT2 and ZT14 showed that NOS RNAi in adult

glia did not affect PDF levels and morphology of the s-LNvs projection patterns. Left, representative confocal images. For visualization purposes only, the contrast of ZT14 images is enhanced. Scale bar, 10 μm. Right, quantification of PDF signal intensity and the area covered by PDF signal within the axonal arbors. Values represent mean ± SD. ****$p < 0.0001$ by one-way ANOVA with Brown-Forsythe and Welch test for multiple comparisons. The numbers of samples are shown in the columns. (B) Average locomotor activity of flies expressing NOS RNAi in perineurial glia (*85G01>NOS RNAi*) and control flies. (n = 30–32 per group) (C) Group average locomotor activity in LD, averaged over 3 days. Rhythmic and arrhythmic flies were separately grouped. The percentage of rhythmic flies in each genotype is shown in Table 2. (D) Diagram of the perineurial glia and subperineurial glia that form the blood-brain barrier in the *Drosophila* adult brain. The perineurial glia contact and brace neighbors through fine extensions, covering the nervous system in a dense layer.

*Drosophila* circadian rhythms. Our results overall suggest that NO exerts temporarily and spatially diverse effects on the *Drosophila* circadian system.

It is rather surprising that the lack of NOS enzyme is not lethal [37] as NO is part of various developmental processes [19–21, 39, 53]. *NOSΔ* mutants are nonetheless strongly arrhythmic in DD and have reduced morning anticipation in LD. Our data suggest that both congenital impairments and lack of NO signaling in adulthood contribute to the behavioral phenotype of the mutants. Axonal terminals of the master pacemaker, s-LNvs, in NOS mutants are profoundly disordered, suggesting the wrong or absent synaptic connections with the downstream partners. However, molecular rhythms in the pacemaker neurons are unaffected. During adulthood, NOS activity in the perineurial glia is required for producing free-running locomotor rhythms but not for maintaining PDF rhythms and structure of the s-LNvs. This finding

**Table 2. Effect of NOS knockdown in glial subpopulations on free-running locomotor rhythms.** The left-most column indicates the expression of drivers are expressed. *n*, number of flies. %R, % of rhythmic flies. Mean periods and rhythmicity of the test groups were compared with those of GAL4-only controls using Student's t-test and chi-square test, respectively.

| | Genotype | Period ± SEM (hr) | Power ± SEM | n | %R |
|---|---|---|---|---|---|
| **UAS-Control** | *NOS-RNAi$^{27725}$/+* | 23.6±0.03 | 168.7±10.72 | 32 | 100 |
| **Astrocyte-Like** | *GMR55B03/+* | 23.2±0.06 | 124.5±8.00 | 31 | 87.1 |
| | *GMR86E01/+* | 23.6±0.14 | 98.7±6.9 | 32 | 84.4 |
| | *alrm/+* | 23.9±0.09 | 107.2±11.79 | 23 | 91.3 |
| | *GMR55B03>NOS-RNAi* | 23.4±0.05* | 177.0±9.17 | 31 | 100 |
| | *GMR86E01>NOS-RNAi* | 23.7±0.04 | 130.1±9.78 | 31 | 93.5 |
| | *alrm>NOS-RNAi* | 24.3±0.06*** | 155.3±10.71 | 23 | 100 |
| **Ensheathing** | *GMR56F03/+* | 23.4±0.05 | 80.8±8.44 | 30 | 80 |
| | *GMR75H03/+* | 23.4±0.05 | 143.0±10.32 | 32 | 90.6 |
| | *GMR56F03>NOS-RNAi* | 23.6±0.03** | 116.8±9.92 | 32 | 75 |
| | *GMR75H03>NOS-RNAi* | 23.5±0.04 | 175.8±10.73 | 32 | 100 |
| **Perineurial** | *GMR85G01/+* | 23.3±0.06 | 92.5±10.37 | 31 | 83.9 |
| | *GMR85G01>NOS-RNAi* | 24.2±0.09**** | 58.8±5.47 | 30 | 30**** |
| **Subperineurial** | *GMR54C07/+* | 23.3±0.05 | 164.2±10.32 | 31 | 96.8 |
| | *GMR54C07>NOS-RNAi* | 24.0±0.06**** | 153.4±11.95 | 32 | 96.9 |
| **Cortex** | *GMR53B07/+* | 23.6±0.06 | 107.9±5.74 | 31 | 80.6 |
| | *GMR54H02/+* | 23.4±0.04 | 120.3±11.31 | 29 | 93.1 |
| | *GMR46H12/+* | 23.6±0.06 | 172.4±11.52 | 29 | 96.6 |
| | *GMR53B07>NOS-RNAi* | 23.6±0.04 | 156.2±11.09 | 31 | 90.3 |
| | *GMR54H02>NOS-RNAi* | 23.5±0.05 | 165.2±12.51 | 30 | 90.0 |
| | *GMR46H12>NOS-RNAi* | 23.7±0.05 | 240.6±6.76 | 30 | 100 |

*$p < 0.05$

**$p < 0.01$

*** $p < 0.001$

****$p<0.0001$.

indicates that NO produced in the perineurial glia is necessary for proper the functioning of circadian locomotor output circuits. Taken together, these results demonstrate that NO signaling is essential for establishing and controlling circadian output circuit.

The functional isoform *dNOS1* shows a circadian variation of its RNA levels throughout the day, which suggest that levels of NO could cycle at least in LD. However, *dNOS* is likely to be regulated by its truncated isoforms in a stage- and cell-type-specific manner, which lays an additional complexity to the regulation of NO production and probably leads to the heterogeneous and context-specific variations of NO. Hyperproduction of NO modulates molecular clockwork, albeit modestly, and is generally detrimental to locomotor rhythms. Therefore, the level and potentially the rhythms of NO production should be tightly controlled in wild-type flies.

In our *NOS RNAi* mini screen, two optic lobe-specific drivers, *GMR79D04* and *GMR85B12*, reduced locomotor rhythmicity in DD. This phenotype was observed when NOS was downregulated constitutively in these cells but not when knockdown was restricted to adulthood. These results reinforce the idea that NO is necessary for a proper establishment of neuronal circuits. A low rhythmicity phenotype caused by NOS knockdown with the pan-neuronal driver *GMR57C10-GAL4* is congruent with the above findings. Intriguingly, however, in addition to the low rhythmicity, *GMR57C10 > NOS RNAi* in adulthood resulted in an extended period. What might be the neuronal subsets that produce NO and regulate period length of locomotor activity? A recent study by the group of Y. Aso and G. Rubin [54], has shown that NO acts as a co-transmitter in a subset of dopaminergic neurons, specifically in some of the PAMs, PPL1s and PPL2abs. It is thus possible that dopamine signaling modulated by NO is involved in the control of the locomotor activity period. It is also noteworthy that NO-mediated signaling has a profound neuromodulatory effect on spinal motor networks and regulates frequency and amplitude of motor activity in various vertebrate species [55–57]. NO-mediated regulation of circadian locomotor output in flies might involve a similar mechanism.

Our DAR4-M staining showed an enrichment of NO in glial cells, including the surface glia. Targeting glial cells leads to the strongest and most persistent phenotype in locomotor activity both for gain- and loss-of-function of NOS. Among glial subpopulations, the perineurial glia appears to be the major site of NOS activity that regulates locomotor rhythms. The importance of glia in circadian rhythms have been recognized, especially those containing the molecular clocks and exert reciprocal communication with the pacemaker neural circuit [58–60]. It has been shown that the perineurial glial cells harbor molecular clocks, which drive daily rhythms in the blood-brain barrier permeability but are not required for locomotor activity rhythms [52]. Our study is the first to identify NO as a signaling molecule produced in glia and mediates part of the role of glia, independently of their molecular clocks, in *Drosophila* circadian rhythms.

It has been shown that in mammals NO mediates light-induced phase-shifts through the cGMP pathway [29]. It is an interesting parallel to note that forced production of NO in the s-LNvs caused phase shift rather than amplitude dampening. Since high levels of NO inhibit E75/UNF dimerization [21] and E75/UNF normally enhances *per* transcription [15], we speculate that NO-induced phase-shift may be partly mediated by the inhibition of E75/UNF heterodimerization. It will be interesting to test this hypothesis in future studies. Mammalian clocks contain E75 homologs REV-ERB α/β, which repress *Bmal1* transcription. Analogous to the notion in flies, NO is thought to decrease REV-ERB α/β activity. Consistently, *in vitro* studies in mammalian cell culture showed that excessive presence of NO increases the production of *Bmal1* mRNA[61]. These findings altogether point out that NO is an evolutionarily conserved regulator of circadian rhythms.

In line with recent studies [7, 62], our research expands the view on the factors that participate in neuronal and molecular mechanisms of circadian rhythmicity. The finding that gaseous messenger NO contributes to the various aspects of circadian rhythmicity emphasizes that non-cell-autonomous, systemic regulation is integral to the circadian circuit operation. Our results set a foundation for future studies addressing the mechanism by which NO signaling modulates the state of the pacemaker circuit and its output.

## Materials and methods

### Fly rearing, crosses, and strains

*Drosophila* were reared at 25˚C on a corn-meal medium under 12 hr:12 hr light-dark (LD) cycles. Two CRISPR deletion mutants *NOSΔter*, *NOSΔall* were kindly provided by O. Schuldiner [21]. The *UAS-macNOS* line was originally generated by H. Krause [19] and provided also by O. Schuldiner. The drivers *GMR57C10, GMR79D04, GMR85B12, GMR33H10, GMR55B03, GMR86E01, alrm-GAL4, GMR56F03, GMR75H03, GMR85G01, GMR54C07, GMR53B07, GMR54H02, GMR46H12* [63], deficiency stock *Df(2L)BSC230*, and *UAS-NOS-RNAi^{56675}* were obtained from Bloomington Stock Center (Indiana, US). The UAS lines *NOS-RNAi^{27725}* and *NOS-RNAi^{108433}* were obtained from the Vienna *Drosophila* Resource Centre (VDRC). The *Clk1982-Gal4* line was provided by N.R. Glossop [64]. The lines *Pdf-Gal4* [65], *Repo-Gal4* [66], *OK107-Gal4* [67], *D52H-Gal4* [68], *GMR-Gal4* [69], *Elav-Gal4* [70], and *R6-Gal4* [71] were described previously. *NOS^{Δ15}* was a gift from P. O'Farrell [37].

### Behavioral assays

The locomotor behavior assay was performed as described previously [14] and data were analyzed using FaasX software [72]. Briefly, male flies were first entrained in 12 h/12 h LD cycles for 4 days and then released in DD for 7–10 days. The flies with power over 20 and width over 2.5h according to the $\chi$2 periodogram analysis were defined as rhythmic. The significance threshold was set to 5%. The $\chi$2 test was used to compare the rhythmicity of the flies, and the Student's *t* test (2-tailed) was used to compare the free-running period. One to four independent experiments were performed for each genotype.

### Immunocytochemistry, microscopy and quantification

The brains were imaged using a Leica SP5 confocal microscope. At least 10 brain hemispheres were subjected to analysis using Image J software (National Institutes of Health). The anti-PER signal was quantified as previously described [14]. Axonal branching patterns of s-LNvs and PDF levels were quantified as previously described [40].

### Nitric oxide visualization and measurements

NO visualization was performed as described in [21] with minor modifications. Brains were dissected in PBS and incubated with 10μM Diaminorhodamine-4M AM (DAR-4M, Sigma-Aldrich) in PBS for 1 h at RT, followed by the fixation for 15 min in PBS containing 4% paraformaldehyde. Immediately after the fixation brains were mounted and imaged. For NO measurements at different times of the day, the procedure was exactly the same with the omission of the fixation step. Long-term NO measurement in *ex-vivo* brain culture was performed as described in [73]. Briefly, brains were dissected on an ice-cold plate in modified Schneider's medium (SMactive) [74] with an addition of 5 mM Bis-Tris (Sigma) and then mounted on a glass-bottom dish (35 mm MatTek petri dish, 20 mm microwell with 0.16/0.19 mm

coverglass). The glass-bottom well was filled with the SMactive medium with 10 μM DAR-4M. Time-lapse imaging was performed at 25˚C and 80%, with images acquired every hour.

### RNA analysis

Total RNA was isolated from adult fly heads using Trizol (Invitrogen) following the manufacturer's protocol. The RNA was reverse-transcribed using oligo(dT) primers, and the resulting cDNAs were quantified using real-time qPCR as previously described [47]. The mRNA levels of *dNOS1* were normalized to those of *elongation factor 1 (Ef1)*.

## Supporting information

**S1 Fig. NOS splice isoforms.** *dNOS1* and presumably *dNOS8* encode functional complete enzyme. The rest isoforms lead to a truncated enzyme. Information is taken from [34]. *dNOS1*-specific primers used are (F) GGC GAG CTT TTC TCC CAG GA, and (R) GAC GAG CCA ATG CTG GAG TC, indicated in red.
(EPS)

**S2 Fig. Upregulation of NO upon macNOS overexpression.** DAR4-M staining of brain expressing *macNOS* in *Pdf-Gal4*. Left, Representative confocal images. Right, comparison of DAR4-M fluorescent levels of a single representative WT and *Pdf > macNOS* brains within the region of s-LNvs.
(EPS)

**S3 Fig. Manipulation of the *NOS* gene does not affect LD locomotor behavior.** Group average locomotor activity of macNOS or NOS-RNAi[27725] expressed with indicated drivers. 4 days of LD are shown.
(TIF)

**S4 Fig. NOS-RNAi efficiency comparison.** NO levels in the brains of flies expressing *NOS-RNAi[27725]* or *NOS-RNAi[108433]* with *Elav-GAL4* were measured using DAR4-M staining. Fluorescence levels were quantified broadly in the region of the central brain, approximately in the area of the central complex. RNAi line 27725 induces a significant reduction of DAR4-M signal. $^{**}p < 0.01$ (Student's test).
(EPS)

**S1 Table. Optic lobe-specific drivers.** Distribution and intensity of the tested generic optic lobe (OL)-specific drivers, taken from Janelia Fly Light project. Original characterization was based on the GFP expression. Expression of all three OL-specific drivers are enriched in the OL. Expression outside the OL is weak and occasional.
(DOCX)

## Acknowledgments

We thank O. Schuldiner and N. Glossop, P. O'Farrell, the Bloomington Drosophila Stock Center and Vienna Drosophila Resource Center for fly stocks. We also thank Fernanda Ceriani for helpful suggestion on the manuscript. We are grateful for our lab members for valuable discussions on this work.

## Author Contributions

**Conceptualization:** Anatoly Kozlov, Emi Nagoshi.

**Data curation:** Anatoly Kozlov, Rafael Koch.

**Formal analysis:** Anatoly Kozlov, Rafael Koch.

**Funding acquisition:** Emi Nagoshi.

**Investigation:** Anatoly Kozlov, Rafael Koch.

**Methodology:** Anatoly Kozlov.

**Supervision:** Emi Nagoshi.

**Visualization:** Anatoly Kozlov, Rafael Koch.

**Writing – original draft:** Anatoly Kozlov, Rafael Koch, Emi Nagoshi.

**Writing – review & editing:** Anatoly Kozlov, Emi Nagoshi.

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
