## [Decision Letter · Decision Letter 0]

19 Aug 2019

Dear Dr Nagoshi,

Thank you very much for submitting your Research Article entitled 'Nitric Oxide Mediates Neuro-Glial Interaction that Shapes Drosophila Circadian Behavior' to PLOS Genetics. I apologize for the delay in completing the review process but I had some difficulty securing reviewers that could adequately cover both the clock and the NO aspects of the work. Your manuscript was fully evaluated at the editorial level and by independent peer reviewers. The reviewers appreciated the attention to an important problem, but raised some substantial concerns about the current manuscript. Based on the reviews, we will not be able to accept this version of the manuscript, but we would be willing to review again a much-revised version. We cannot, of course, promise publication at that time.

If you decide to revise the manuscript for further consideration at PLOS Genetics, please aim to resubmit within the next 60 days, unless it will take extra time to address the concerns of the reviewers, in which case we would appreciate an expected resubmission date by email to plosgenetics@plos.org.

[LINK]

We are sorry that we cannot be more positive about your manuscript at this stage. Please do not hesitate to contact us if you have any concerns or questions.

Yours sincerely,

John Ewer

Associate Editor

PLOS Genetics

Gregory P. Copenhaver

Editor-in-Chief

PLOS Genetics

Reviewer #2: This study is aimed at understanding the role for Nitric Oxide Synthase (NOS) in the control of circadian behavioral rhythms in Drosophila. Partial loss of rhythmicity is observed with both mutants tested, and with RNAi driven by different GAL4 drivers. The authors propose that glia is a likely important source of NO. This would establish a novel role for glia in the regulation of circadian behavior. Interestingly also, dorsal projections of the sLNvs, the main circadian pacemaker neurons, are defective in NOS mutants, and this might explain the partial loss of rhythmicity. Overall, this study is innovative, but the authors need to better support their claims. For example, it would be important to make sure that the phenotypes indeed map to the nos gene. Also, the model that NO would work through E75/UNF is very exciting, but direct evidence is lacking.

Major comments:

1) I am puzzled by the fact that the complete deletion of the nos gene has a significantly weaker phenotype than the partial deletion, both in LD and DD. Why is it so? Is the partial deletion predicted to eliminate all NOS isoforms? If so, the two alleles should behave similarly. Could genetic background be an issue? I presume both mutants were generated from the same genetic background, so the absence of complementation in nos-ter/nos-all flies could be the result of a secondary mutation. Unfortunately, rescue with the GAL4/UAS system is not an option since overexpression causes arrhythmicity as well. Thus, I would suggest to validate the observations with backcrossing (unless this was actually done) and by crossing the mutants to deficiencies to map the phenotype to the chromosomal region of NOS.

2) Is the morning peak of activity also eliminated or reduced with RNAi?

3) The complete deletion results in 50% of flies being arrhythmic, while the rhythmic flies seem quite normal based on power. Are the average activity plots an average of both rhythmic an arrythmic flies? What happens to the morning anticipation if the rhythmic and arrythmic flies are separated? Is it completely missing in arrhythmic flies, but present in rhythmic flies? Does this correlate with the severity of the defect in synaptic projections? It would be worth staining brains of rhythmic and arrhythmic flies to see if there is a correlation with the severity of the sLNv projection defects.

4) There is clear defect in the sLNv projections, but I have a hard time seeing the rhythms in projection complexity in the control flies (fig3). Quantifications would be helpful here. Also, length was measured, but is thickness also affected. Number of branches?

5) I presume that on figure3A pdf-gal4 was used, as in figure S2. This should be indicated. But what are the more or less circular structures seen in the nos mutants, particularly in nos-ter? Some are quite big. Again, it seems overall severity of the phenotype is more severe in nos-ter than nos-all. This correlates with behavior phenotypes, but not with the severity of the genetic lesion.

6) The model that NOS would work through UNF/E75 is very interesting. Could it be tested through genetic epistasis, or with double heterozygotes? Or is there a reporter for UNF/E75 that could be used in sLNvs?

7) That the UNF-RNAi and nos mutant projections phenotypes are similar is not obvious to me. The projection lengths in UNF RNAi appear shorter than control, while in nos mutants they are longer. Moreover, it seems the projections are thinner and less structured in nos mutant than UNF. Perhaps this is reflecting the quality of the staining in the two experiments, but again, some form of quantification would be important.

8) Lines 221-223 are not very clear, but after reading lines 331-332 in the discussion, I see what the authors meant. However, if UNF/E75 increase per transcription, and if overexpression of NOS increases UNF/E75 dimerization, is this not supposed to advance PER accumulation and shorten period, rather than delaying PER accumulation? Increased per transcription leads to short period (Kadener 2008), and the senior author's group showed that losses of UNF or E75 in adult lengthen period. Therefore, the NOS overexpression phenotype does not seem to fit well with the idea that it works though UNF/E75. As mentioned above, it would be important to gather additional support for this idea.

9) I would suggest adding the timGAL4/repoGAL80 combination to the RNAi and overexpression studies.

10) Why does pan-neuronal RNAi cause a long period phenotype, but not the complete knock-out? Could this be an off-target effect? A possible way to check this is to perform the RNAi experiment in a nos null background. There should be no period lengthening there. A second RNAi line would be an alternative.

Minor:

1) Line 233-234: Is molecular cycle phase in the sLNvs really misaligned with the rest of the circadian neural network? This would be interesting to know

2) Nos mRNA rhythms should be tested under DD too.

3) Is there a reference to support the idea that the R57C10 driver is stronger than elaV (line 244-45)?

4) In the adult specific knock-out, the GMR84B12 control was very arrhythmic, so the level of arrhythmicity in the RNAi cross is not meaningful. It might be best to remove these flies from the table.

5) Figure 2B, legend. A graph is shown, not an histogram as indicated in the legend.

Reviewer's Responses to Questions

**Comments to the Authors:**

Reviewer #1: Comments to Kozlov and Nagoshi.

The manuscript by Kozlov and Nagoshi describes the involvement of the signaling molecule nitric oxide (NO) in the fly brain, and particularly, its relevance to different outputs of the small LNvs. The authors characterize rhythmic locomotor activity and structural plasticity in two deletion mutants and propose that the structural aberrations trigger the behavioral phenotypes without affecting the molecular clock. Such effects appeared not to be cell-autonomous, and the authors set out to look for the source of the NO signal through Gal4 directed overexpression of a constitutive version of the NO synthase or RNAi-mediated downregulation. As the authors recognize in the text, NO plays a role within the mammalian SCN, and it would be interesting to investigate its relevance to the Drosophila circadian network further. However, despite the inherent difficulty of analyzing the impact of a diffusible signal in vivo, there are several interpretations that need to be toned down and further confirmed.

Specific comments.

In Figure 1, panels B and C. Statistical comparisons of 3 genotypes -even if carried out pairwise- cannot be performed by a Student´s t test. Please revise the statistical analysis throughout the manuscript.

Figure 3. Panel A. Both mutants show aberrant projections; however, despite both are loss of function alleles the phenotypes vary a great deal: in addition to some miss-routing of specific terminals and the presence of secondary or tertiary neurites, the integrity of the membrane itself appears damaged, particularly so in the representative examples shown for NOS(delta)all. Is there any explanation to such a difference when both mutations apparently give rise to a similar reduction (Fig. 1C)? Also, have the authors quantitate PDF levels in those samples? That would be an important control to establish the link between the morphology of the terminals and rhythmic behavior.

Likewise, the authors mention that loss of NO signaling during development could affect the correct wiring and thus be responsible for the defective behavior, but this possibility was not tested (Given the relevance of this claim to the take home message perhaps it should be).

Another interesting observation is described in Fig 2. In terms of the behavioral phenotypes, aside from the startle effect Panel B shows that mutants mostly lack the anticipatory activity to lights off (the evening peak), as opposed to lights on. How can this be reconciled with a defective output of the neurons controlling the M peak? Please explain.

The authors test whether the transcription factor UNF would be involved in the defective NO response that triggers the structural defects; however chronic expression of UNF RNAi in the LNvs gives rise to highly arborized terminals, this phenotype does not resemble at all the ones described in the mutant. How can this be reconciled? Additional, more specific, comments are included below.

If PDF neurons do not express NOS (according to Table I), how relevant is that macnos expression triggers a clear phenotype in the molecular clock (Fig 4D)? Could this be an indirect effect of affecting PDF levels? How do the arborizations look under these conditions?

To identify potential sites of NO production macNos OX and Nos KD was attempted in different groups of cells. Interestingly, they found that both treatments trigger a behavioral phenotype when expressed under a panglial promoter (Repo), a pancircadian promoter (tim), as well as a panneuronal one (GMR57C10). How can this be explained? In addition, immunohistochemistry should be performed –ideally in chronic and adult- specific fashion) to corroborate the link between the behavioral phenotypes and the complexity of the sLNv projections.

There is room for improvement in figure legends (all, in the main text as well as the supplementary one) and in the figures themselves. Figure legends do not provide critical information to assess the experiments; labeling within the figures is also scarce.

Additional comments.

Fig. 1 panel C. The y axis is such that the curves describing DARM staining in both mutants is difficult to see. Could that initial segment be expanded?

Legend to Supp Table I. Page 24, line 661, please revise ….¨in compare with¨

Figure 2. In panel A, mutant activity profiles are compared to w1118 control, while in panel B are compared to CS. What is the genetic background of these mutants? Please clarify or include the proper genetic controls. There are several phenotypes associated to the knock-down NOS alleles, a reduced startle effect to lights on/off, a decreased evening anticipation, and even reduced overall activity. Shouldn´t some of the phenotypes (i.e. quantitation) be included in this main figure?

The grey shading is missing in the first day, before lights on.

Fig 2 legend. N is usually employed to refer to the number of experiments performed, and n to the number of animals tested. I would suggest changing to lowercase, and describe in the legend how many experiments were carried out in each case.

Supplementary Fig 2. The figure includes ¨representative images¨ at ZT2, presumably from 3 independent animals, which should be detailed in the figure legend. Also, is it possible that the labels are mistaken? The ¨control¨-labeled ones clearly do not look normal projections at ZT2. The authors could do a better job at describing the terminals (complexity/degree of arborization using Sholl analysis or the Image J plug in; the length of principal neurites, etc).

Figure 4, panel C. The image shows a single LNv stained for DAR4M above background. Was this always the case (a single small LNv/hemisphere)? That would indeed be interesting, as not too many differences within the cluster have been described. At what timepoint was the staining performed? Panel D, left. Please include labels indicating the times each image refers to.

Reviewer #2: This study is aimed at understanding the role for Nitric Oxide Synthase (NOS) in the control of circadian behavioral rhythms in Drosophila. Partial loss of rhythmicity is observed with both mutants tested, and with RNAi driven by different GAL4 drivers. The authors propose that glia is a likely important source of NO. This would establish a novel role for glia in the regulation of circadian behavior. Interestingly also, dorsal projections of the sLNvs, the main circadian pacemaker neurons, are defective in NOS mutants, and this might explain the partial loss of rhythmicity. Overall, this study is innovative, but the authors need to better support their claims. For example, it would be important to make sure that the phenotypes indeed map to the nos gene. Also, the model that NO would work through E75/UNF is very exciting, but direct evidence is lacking.

Major comments:

1) I am puzzled by the fact that the complete deletion of the nos gene has a significantly weaker phenotype than the partial deletion, both in LD and DD. Why is it so? Is the partial deletion predicted to eliminate all NOS isoforms? If so, the two alleles should behave similarly. Could genetic background be an issue? I presume both mutants were generated from the same genetic background, so the absence of complementation in nos-ter/nos-all flies could be the result of a secondary mutation. Unfortunately, rescue with the GAL4/UAS system is not an option since overexpression causes arrhythmicity as well. Thus, I would suggest to validate the observations with backcrossing (unless this was actually done) and by crossing the mutants to deficiencies to map the phenotype to the chromosomal region of NOS.

2) Is the morning peak of activity also eliminated or reduced with RNAi?

3) The complete deletion results in 50% of flies being arrhythmic, while the rhythmic flies seem quite normal based on power. Are the average activity plots an average of both rhythmic an arrythmic flies? What happens to the morning anticipation if the rhythmic and arrythmic flies are separated? Is it completely missing in arrhythmic flies, but present in rhythmic flies? Does this correlate with the severity of the defect in synaptic projections? It would be worth staining brains of rhythmic and arrhythmic flies to see if there is a correlation with the severity of the sLNv projection defects.

4) There is clear defect in the sLNv projections, but I have a hard time seeing the rhythms in projection complexity in the control flies (fig3). Quantifications would be helpful here. Also, length was measured, but is thickness also affected. Number of branches?

5) I presume that on figure3A pdf-gal4 was used, as in figure S2. This should be indicated. But what are the more or less circular structures seen in the nos mutants, particularly in nos-ter? Some are quite big. Again, it seems overall severity of the phenotype is more severe in nos-ter than nos-all. This correlates with behavior phenotypes, but not with the severity of the genetic lesion.

6) The model that NOS would work through UNF/E75 is very interesting. Could it be tested through genetic epistasis, or with double heterozygotes? Or is there a reporter for UNF/E75 that could be used in sLNvs?

7) That the UNF-RNAi and nos mutant projections phenotypes are similar is not obvious to me. The projection lengths in UNF RNAi appear shorter than control, while in nos mutants they are longer. Moreover, it seems the projections are thinner and less structured in nos mutant than UNF. Perhaps this is reflecting the quality of the staining in the two experiments, but again, some form of quantification would be important.

8) Lines 221-223 are not very clear, but after reading lines 331-332 in the discussion, I see what the authors meant. However, if UNF/E75 increase per transcription, and if overexpression of NOS increases UNF/E75 dimerization, is this not supposed to advance PER accumulation and shorten period, rather than delaying PER accumulation? Increased per transcription leads to short period (Kadener 2008), and the senior author’s group showed that losses of UNF or E75 in adult lengthen period. Therefore, the NOS overexpression phenotype does not seem to fit well with the idea that it works though UNF/E75. As mentioned above, it would be important to gather additional support for this idea.

9) I would suggest adding the timGAL4/repoGAL80 combination to the RNAi and overexpression studies.

10) Why does pan-neuronal RNAi cause a long period phenotype, but not the complete knock-out? Could this be an off-target effect? A possible way to check this is to perform the RNAi experiment in a nos null background. There should be no period lengthening there. A second RNAi line would be an alternative.

Minor:

1) Line 233-234: Is molecular cycle phase in the sLNvs really misaligned with the rest of the circadian neural network? This would be interesting to know

2) Nos mRNA rhythms should be tested under DD too.

3) Is there a reference to support the idea that the R57C10 driver is stronger than elaV (line 244-45)?

4) In the adult specific knock-out, the GMR84B12 control was very arrhythmic, so the level of arrhythmicity in the RNAi cross is not meaningful. It might be best to remove these flies from the table.

5) Figure 2B, legend. A graph is shown, not an histogram as indicated in the legend.

Reviewer #3: Kozlov and Nagoshi investigated role of nitric oxide synthase (NOS) in regulation of circadian behavior in Drosophila. First they found mutants lacking NOS have defect in circadian locomotion. The same mutant also showed developmental defect of the key pacemaker neurons. In cultured brains, glial cells exhibited strong DAR4-M signals, dye sensor that become fluorescent after reacting to NO. Together with glia specific knockdown of NOS in adult flies, they concluded that NOS is active in adult brains and is an essential mediator of neuro-glial interaction in circadian system. These findings can be of general interests, but need to be validated with additional control experiments.

Major

1) In which glial cell type NOS is expressed? Resolution of genetic dissection is insufficient to gain mechanistic insights about interaction between specific type of glia and neurons. Kremer et al., GLIA 2017 (https://doi.org/10.1002/glia.23115) reported many driver lines for distinct types of glia. Authors should address which type of glia express NOS by RNA-Seq using these driver lines or FISH experiments. Cite papers if there is any published RNA-seq or quantitative PCR data for supporting NOS expression in glia. Davis et al., reported that there is no NOS expression in glia in the optic lobe (http://dx.doi.org/10.1101/385476).

2) Following (1), validate that NOS proteins express in defined glial cell type by immunohistochemistry, and NOS immunoreactity can be reduced by NOS RNAi knockdown. Antibodies may not distinguish splicing isoform of NOS, but it depends on which splicing isoforms are expressed. Also it is important to examine where in the glial cells NOS localize.

3) Interpretation of DAR4-M signals requires additional control for distribution of dye itself. Given the architecture of insect brains, DAR4-M needs to pass through glia to penetrate to neuropils. Thus, seemingly glia specific signals of DAR4-M could be because DAR4-M was stuck in glia and did not penetrate inside neuropils. DAR4-M fluorescent signals need to be normalized by local concentration of DAR4-M. After loading DAR4-M, fix brains and exposing them to chemical donor of NO such as NOC-7 to examine loading of DAR4-M.

4) Page 8, line 200: “The signal was particularly high within and around the central complex and in the optic lobe” Provide more detail anatomical description and images.

5) Regulski and Tully reported that NOS in Drosophila is calcium/calmodulin dependent (https://doi.org/10.1073/pnas.92.20.9072). Thus it is important to consider how NOS could be activated in glia. Is calcium dynamics in cultured brains comparable to that of physiological state? Is it possible that DAR4-M fluorescent signals in glia could be due to NO released from neurons?

6) If glial NOS is primary source of NO, NOS-RNAi with repo-GAL4 but not 57C10-GAL4 should reduce DAR4-M fluorescent signals in glia.

**Have all data underlying the figures and results presented in the manuscript been provided?**

Reviewer #1: Yes

Reviewer #2: Yes

Reviewer #3: Yes

PLOS authors have the option to publish the peer review history of their article (what does this mean?). If published, this will include your full peer review and any attached files.

Reviewer #1: No

Reviewer #2: No

Reviewer #3: No

---

## [Decision Letter · Decision Letter 1]

12 Mar 2020

Dear Dr Nagoshi,

Thank you very much for submitting a revised version of your Research Article entitled 'Nitric Oxide Mediates Neuro-Glial Interaction that Shapes Drosophila Circadian Behavior' to PLOS Genetics. Your manuscript was fully evaluated at the editorial level and by independent peer reviewers. The reviewers appreciated the attention to an important topic but identified some aspects of the manuscript that should be improved.

We therefore ask you to modify the manuscript according to the review recommendations before we can consider your manuscript for acceptance. Your revisions should address the specific points made by each reviewer.

1) Provide a detailed list of your responses to the review comments and a description of the changes you have made in the manuscript. We also ask that you please include a version in which any text that is added or changed is in a different color font. In the current revised version new text was not highlighted, making it harder for reviewers to identify the changes.

[LINK]

Yours sincerely,

John Ewer

Associate Editor

PLOS Genetics

Gregory Copenhaver

Editor-in-Chief

PLOS Genetics

Reviewer's Responses to Questions

**Comments to the Authors:**

Reviewer #1: Overall the manuscript has improved as a result of the revision process, despite some of the requested experiments were not carried out. Given that a major claim is that NO plays a crucial role during circuit establishment/refinement, I am puzzled by the fact that the most obvious test to this hypothesis was not performed; that is, to knockdown Nos ONLY during development, combining the PG specific driver or even timGal4 with tubGal80ts, to enable Gal4 activity during that stage. Does this combination also lead to lethality? I would be surprised.

In Figure 6, have the authors described the axonal area employing PDF staining? As mentioned earlier, this is far from ideal since PDF containing DCV not necessarily describe the complexity of the arborizations. Moreover, it has already been established that affecting the clock in Repo+ glial cells impairs circadian remodeling of the sLNv terminals without affecting PDF cycling/levels. In the example included in the figure, controls show smaller differences between the time points analyzed....

Some use of language still needs to be revised:

line 170... branching pattern was highly disordered and fuzzy

line 320 ...have an ¨utterly wrong¨ shape

line 324 ...NO produced in the perineurial glia is necessary for proper the functioning

Reviewer #2: The authors have addressed all my comments. The main conclusions of the paper are now more solidly supported, particularly with the addition of a new null nos allele. The partial loss of rhythmicity is clearly reproducible with multiple alleles. The structural defects in sLNvs are striking, and a role for nos in glia is convincing. However, the connection with E75/UNF was entirely removed from the manuscript, and sLNv structural defects do not explain the loss of rhythmicity. Therefore, the study is quite descriptive in nature, given the absence of a mechanism for the loss of rhythmicity. This said, establishing that NO plays a role in circadian rhythmicity, and that perineurial glia is an important locus of NO production and thus a modulator of the amplitude of circadian rhythms is an innovative and important discovery that will certainly attract the attention of chronobiologist and neuroscientist.

Reviewer #3: The new data about perineurial glia is valuable addition. However, authors need more direct evidence to support their conclusion “we identify the perineurial glia, one of the two glial subtypes that form the blood-brain barrier, as the major source of NO that regulates circadian locomotor output.”. I request either one of two experiments below.

1) Show a direct evidence that NOS is expressed in perineurial glia by immunohistochemistry, fluorescence in situ hybridization or RNA-seq. Behavioral phenotype with NOS-RNAi isn’t a direct evidence of NOS expression.

2) Show an evidence that DAR4M signals in glia is dependent on NOS in glia. DAR4M signal in glia isn’t necessary an outcome of reacting to NO produced in glia, because NO from neurons may diffuse and react to DAR4M loaded in glia. Express GFP and NOS-RNAi/control-RNAi in glia by 85G01-GAL4 and measure reduction of DAR4M signals in perineurial glia.

**Have all data underlying the figures and results presented in the manuscript been provided?**

Reviewer #1: Yes

Reviewer #2: Yes

Reviewer #3: Yes

PLOS authors have the option to publish the peer review history of their article (what does this mean?). If published, this will include your full peer review and any attached files.

Reviewer #1: No

Reviewer #2: No

Reviewer #3: No

---

## [Editor Report · Decision Letter 2]

22 May 2020

Dear Dr Nagoshi,

Thank you very much for submitting your re-revised Research Article entitled 'Nitric Oxide Mediates Neuro-Glial Interaction that Shapes Drosophila Circadian Behavior' to PLOS Genetics. We appreciate the effort you made to revise the manuscript following our suggestions. Before we can press the "accept" button we must however ask you to improve Figures 4D (DAR4M panel) and 6A, as the staining pattern is really quite difficult to see. We would suggest you try inverting the color scheme, making these panels black (staining) on white (background). 

In addition we ask that you to upload a Striking Image with a corresponding caption to accompany your manuscript if one is available (either a new image or an existing one from within your manuscript). If this image is judged to be suitable, it may be featured on our website. Images should ideally be high resolution, eye-catching, single panel square images. For examples, please browse our archive. If your image is from someone other than yourself, please ensure that the artist has read and agreed to the terms and conditions of the Creative Commons Attribution License. Note: we cannot publish copyrighted images.

[LINK]

Yours sincerely,

John Ewer

Associate Editor

PLOS Genetics

Gregory P. Copenhaver

Editor-in-Chief

PLOS Genetics

---

## [Editor Report · Decision Letter 3]

5 Jun 2020

Dear Dr Nagoshi,

Thank you for making the final revisions to your manuscript; the figures are indeed much improved. As a result, we are pleased to inform you that your manuscript entitled "Nitric Oxide Mediates Neuro-Glial Interaction that Shapes Drosophila Circadian Behavior" has now been editorially accepted for publication in PLOS Genetics. Congratulations!

Yours sincerely,

John Ewer

Associate Editor

PLOS Genetics

Gregory P. Copenhaver

Editor-in-Chief

PLOS Genetics

Comments from the reviewers (if applicable):

**Data Deposition**

http://datadryad.org/submit?journalID=pgenetics&manu=PGENETICS-D-19-01150R3

**Press Queries**

---

## [Editor Report · Acceptance letter]

22 Jun 2020

PGENETICS-D-19-01150R3 

Nitric Oxide Mediates Neuro-Glial Interaction that Shapes Drosophila Circadian Behavior 

Dear Dr Nagoshi, 

We are pleased to inform you that your manuscript entitled "Nitric Oxide Mediates Neuro-Glial Interaction that Shapes Drosophila Circadian Behavior" has been formally accepted for publication in PLOS Genetics! Your manuscript is now with our production department and you will be notified of the publication date in due course.

With kind regards,

Jason Norris

PLOS Genetics

On behalf of:
